# Structural basis for the biosynthesis of lovastatin

Jialiang Wang [1,4], Jingdan Liang[1,4], Lu Chen[1], Wei Zhang[1], Liangliang Kong[2], Chao Peng [2], Chen Su[2], Yi Tang[3], Zixin Deng [1✉] & Zhijun Wang [1✉]

Statins are effective cholesterol-lowering drugs. Lovastatin, one of the precursors of statins, is formed from dihydromonacolin L (DML), which is synthesized by lovastatin nonaketide synthase (LovB), with the assistance of a separate *trans*-acting enoyl reductase (LovC). A full DML synthesis comprises 8 polyketide synthetic cycles with about 35 steps. The assembling of the LovB–LovC complex, and the structural basis for the iterative and yet permutative functions of the megasynthase have remained a mystery. Here, we present the cryo-EM structures of the LovB–LovC complex at 3.60 Å and the core LovB at 2.91 Å resolution. The domain organization of LovB is an X-shaped face-to-face dimer containing eight connected domains. The binding of LovC laterally to the malonyl-acetyl transferase domain allows the completion of a L-shaped catalytic chamber consisting of six active domains. This architecture and the structural details of the megasynthase provide the basis for the processing of the intermediates by the individual catalytic domains. The detailed architectural model provides structural insights that may enable the re-engineering of the megasynthase for the generation of new statins.

[1] State Key Laboratory of Microbial Metabolism and School of Life Science & Biotechnology, Shanghai Jiao Tong University, Shanghai, China. [2] National Facility for Protein Science in Shanghai, Shanghai, China. [3] Department of Chemical and Biomolecular Engineering and Department of Chemistry and Biochemistry, University of California, Los Angeles, CA, USA. [4] These authors contributed equally: Jialiang Wang, Jingdan Liang. ✉email: zxdeng@sjtu.edu.cn; wangzhijun@sjtu.edu.cn

Statins are inhibitors of hydroxymethylglutaryl-coenzyme reductase (HMG-CoA), which converts HMG-CoA to mevalonate, the rate-limiting step in cholesterol biosynthesis. This activity enables the medicinal use of statins to treat hypercholesterolemia and potentially to reduce mortality in multiple cancer types[1]. Lovastatin is a precursor of the multi-billion sold semisynthetic statins. It is biosynthesized in the filamentous fungus *Aspergillus terreus*[2–5]. The first isolable intermediate of the lovastatin biosynthetic pathway, dihydromonacolin L (DML), is constructed by the highly reducing iterative polyketide synthase (HR-iPKS) LovB, which partners with LovC, an enoyl reductase that acts *in trans*[6,7].

LovB is a representative polyketide synthase that shares a common architecture and domain structures with animal and bacterial PKSs, which synthesize chemically diverse drugs and bioactive compounds[8] (Supplementary Fig. 1a). In LovB, the β-ketoacyl synthase (KS), malonyl-acetyl transferase (MAT), dehydratase (DH), methyltransferase (CMeT), acyl carrier protein (ACP), and ketoreductase (KR) domains are active, except for the enoyl reductase (ER) domain. LovB terminates with a condensation (CON) domain commonly found in nonribosomal peptide synthetases (Supplementary Fig. 1b).

In DML biosynthesis by LovB, the initiation and elongation of the intermediate chain are carried out by the KS, MAT, and ACP domains. ß-Ketoacyl modification cycles are also repeated, but the tailoring domain usage during each iteration is highly programmed and permutative (Supplementary Fig. 1c). The combination of the KR–DH domains functions in iterations 1, 2, and 5, while the combination of the KR–DH–ER domains functions in iterations 4 and 6. The full domain usage of CMeT–KR–DH–ER occurs only once at iteration 3. The DH domain is then omitted, with only the KR domain used in the final two iterations 7 and 8. A Diels-Alder reaction is proposed to take place on the triene chain after iteration 5[7]. The structural basis for programming a single set of catalytic domains in the megasynthase differently during each iteration has remained a mystery for LovB.

Excellent domain swapping experimental results and biochemical assays of an isolated enoyl reductase from the Cox and Townsend labs have demonstrated, that the individual catalytic modifying domains themselves possess selectivity for specific substrates[9,10]. The CMeT domain, which catalyzes methylation in polyketide formation for citrinin, functions as a gatekeeper[11]. The starter unit selection carried out by the SAT domain for PksA also contribute to the selectivity[12]. These observations led to the proposal that in the programming of iterative PKSs, ultimate control resides in the structure of the protein and the recognition of structurally ever-changing substrates[9].

Current structural knowledge of HR-iPKS is derived from a bacterial type I PKS module[13,14], a hybrid MAS-like PKS[15], the mammalian FAS (mFAS)[16,17] and several individual PKS domains[18], which is insufficient for the understanding of lovastatin biosynthesis and the programming mechanism for HR-iPKSs. Therefore, key structural questions regarding topics such as the assembly of the whole enzyme, the architecture of the catalytic chamber, and the detailed constitution of individual catalytic tunnels need to be solved. Particularly in the biosynthesis of DML, the ER domain plays a key role in ensuring proper programming of the PKS, and such *trans*-acting ERs have been reported across various fungal PKS enzymes as a tactic in nature to diversify polyketides[19]. The site of interaction between the ER domain and the LovB PKS is unknown. These questions represent a barrier to full understanding of the catalytic cycles.

In this work, we determine the structures of LovB at 2.91 Å and the LovBC complex at 3.60 Å resolution using cryo-electron microscopy. The structure of LovB adopts an X-shaped dimer architecture, and the LovBC complex reveals the position of LovC

which binds to the MAT domain of LovB, forming two complete L-shaped catalytic chambers. Mutational analysis of the LovB–C interface confirms the essential role of the catalytic chamber integrity for the production of DML. Together, our observation provides the structural basis for the iterative yet programmed biosynthesis of lovastatin.

## Results

**Overall architecture of the LovBC complex.** We purified the full-length His-tagged LovB and LovC separately using a nickel column (Supplementary Fig. 2a, b). We then mixed LovB and LovC in a stoichiometric ratio of 1:1.2. The complex was then further purified using a final size exclusion column to remove the extra LovC (Supplementary Fig. 2c). Treatment of the protein sample using 1 mM DSS crosslinker before the size exclusion step helped reduce the monomeric LovB contaminant (Supplementary Fig. 2d). After the SEC step, three cofactors were re-added into the LovBC solution for cryo-EM sample preparation, ensuring that the particles were homogeneous enough for structure determination (Supplementary Fig. 3). None of the cofactors except NADP$^+$ was observed in the final models. High-quality LovBC cryo-EM particles were obtained only by adding 2 mM of each of three cofactors (Mal-CoA, NADPH, and SAM) to the sample with the incubation for 1 h but in the absence of LovG (the product releasing thioesterase). The gain of high-quality particles could be due to that the majority of the protein complex was pushed to a conformation representing the final stage of the DML synthesis just before release. We collected cryo-EM images with a K2 Summit direct electron detector equipped on a Titan Krios electron microscope, and RELION was used for image processing[20–25]. Rounds of 2D and 3D classification for particle selection and refinements were performed. Finally, the maps of overall 2.91 Å for LovB and 3.60 Å resolution for the LovBC complex were reconstructed (Supplementary Figs. 4 and 5). The resolution of the maps enabled us to reliably assign the individual domains, dissect their linker junctions and finally build the atomic models of LovB (Supplementary Fig. 6).

The LovBC complex adopts an X shape in the front view with two wings. Starting from the lower region, the KS and MAT domains are connected by the linker domain (LD), with the separate ER (LovC) domain interacting with the MAT domain (Fig. 1a, b). The MAT domain is linked to the upper region, which begins with the DH domain, followed by the intact CMeT domain, protruding from the relatively planar body of the whole complex. Then, the truncated ψKR domain is linked to the nonfunctional ψER domain, and the connected KR domain finally completes the upper tailoring region. The ACP and tethered CON domains were not solved, possibly due to the inherent flexibility of the ACP domain. The height of the complex is approximately 152 Å. LovB alone is ~176 Å wide, while the LovBC complex is ~294 Å wide. The thickness of the complex is ~95 Å. The two monomers contact each other with an approximately a 6197 Å$^2$ interface, which was mainly contributed by the KS, ψER, and DH domains, with buried surface areas of 2963 Å$^2$, 1545 Å$^2$, and 1250 Å$^2$, respectively. The post-MAT wing-junction linkers (Supplementary Fig. 7) mediate the contact of the upper wing with the lower wing. Contact between the KS and DH domains with a surface area of 304.78 Å$^2$ was also observed. Rounds of 3D classification generated two structures with slight differences in domain angles (≈0.4°), suggesting that the dynamic mobility of the whole structure is minimal (Supplementary Fig. 8).

The adoption of a pseudo-twofold face-to-face symmetry of LovB, in connection with LovC, creates two L-shaped catalytic chambers. The chamber is chimeric, with the KS, and MAT

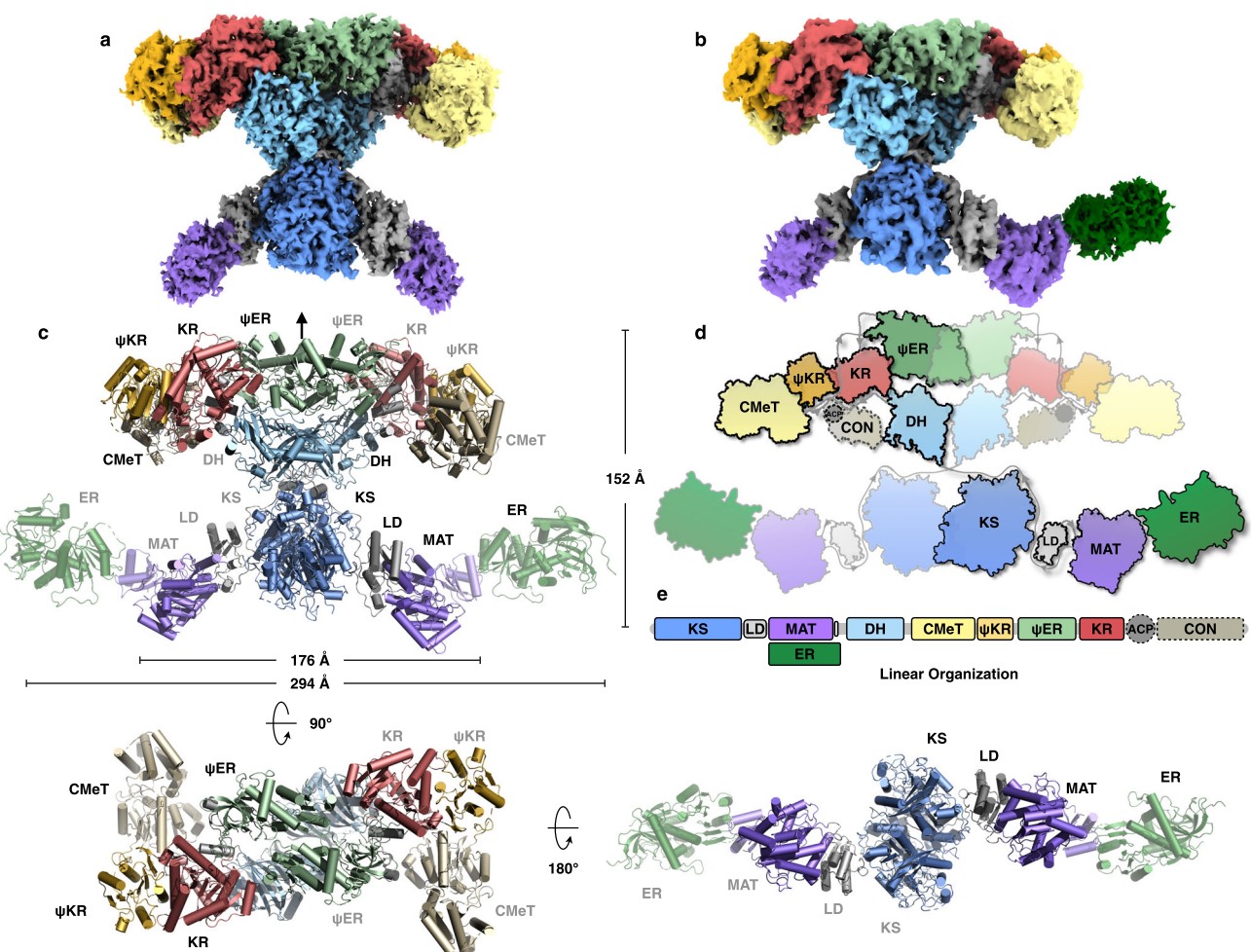

**Fig. 1 Overall architecture of the LovBC complex.** Cryo-EM density maps of LovB (**a**) at 2.91 Å and the LovBC complex (**b**) at 3.60 Å resolution with each domain colored uniquely. **c** The atomic model of the LovBC complex with the dimensions indicated is shown in front, top, and bottom views. The pseudo-twofold symmetry axis is indicated by an arrow. **d** Schematic diagram of the domain arrangements illustrating the X shape of LovBC. **e** Linear domain organization of the LovBC complex. The unresolved ACP and CON domains are bordered by dotted lines.

domains coming from one chain, and the DH, CMeT, ψKR, ψER, and KR domains from another (Fig. 1c–e).

**Structural analysis of LovB enzymatic domains.** The dimeric KS–LD–MAT domains adopt a linear extended conformation (Fig. 2a), similar to the homologous mFAS and DEBS M3. Briefly, the MAT domain is slightly rotated relative to KS. They are connected by the 3α2β-fold linker domain (LD). The post-MAT linker can be divided into two parts. The lower part, together with the LD, play the structural roles in fixing position between KS and MAT. The upper part, defines the relative organization of the DH domains, and hence arranges the dimers into the face-to-face X conformation. The linker mediates the contact between the upper and lower wings and contributes to the fixative assembly of the complex as well (Supplementary Fig. 7), also no other significant conformation of LovB was detected through cryo-EM data processing, in contrast to mFAS, which has its lower wing rotated relative to the upper[26]. The α, β-hydrolase core domain and ferredoxin-like subdomain contained MAT domain have conserved S656, R681, and H763 active site residues, compared with DEBS and mFAS (r.m.s deviation of 1.89 Å and 1.67 Å, respectively). The KS domain adopts an αβαβα structure and contains the conserved C181, H320, and H367 active site residues with r.m.s deviation of 1.57 Å and 1.81 Å, respectively, to DEBS and mFAS (Supplementary Fig. 13).

Despite the common fold shared by the KS domains, the substrate tunnels in the KS domains of LovB show differences from those of mFAS and DEBS, highlighted by the disconnection of the PPant pocket from the acyl pocket (Fig. 2b), possibly due to the hydrophobic interaction of the residue F436 with M132. Although the acyl pocket tunnel is disconnected from the PP pocket tunnel observed here, it might represent a conformational state in the absence of an intermediate in the tunnel. The hydrophobic interacting residue pair (F436 and M132) could potentially function as part of a gate[27,28], that dynamically controls substrate access to the active site, prevents solvent access to specific regions of the protein, or synchronizes processes occurring in distant parts of the megasynthase. This truncated tunnel and its detailed constitutional surface may underpin the specific recognition of the relatively short acyl intermediates during the polyketide elongation cycles in DML synthesis.

The NADPH-dependent KR domain adopts the typical Rossmann fold, which belongs to the SDRs (short-chain dehydrogenases/reductases) family (Fig. 3a). The NADPH binding pocket with bound NADP+ nicotinamide ring can be clearly identified. The active site residues S2294 and Y2307 are conserved and the substrate tunnel travels along them. The structure is closely related to those of mFAS and Amp module 11 of modPKS (r.m.s deviation of 1.57 Å and 1.27 Å, respectively). While the substrate entry groove was narrower compared with

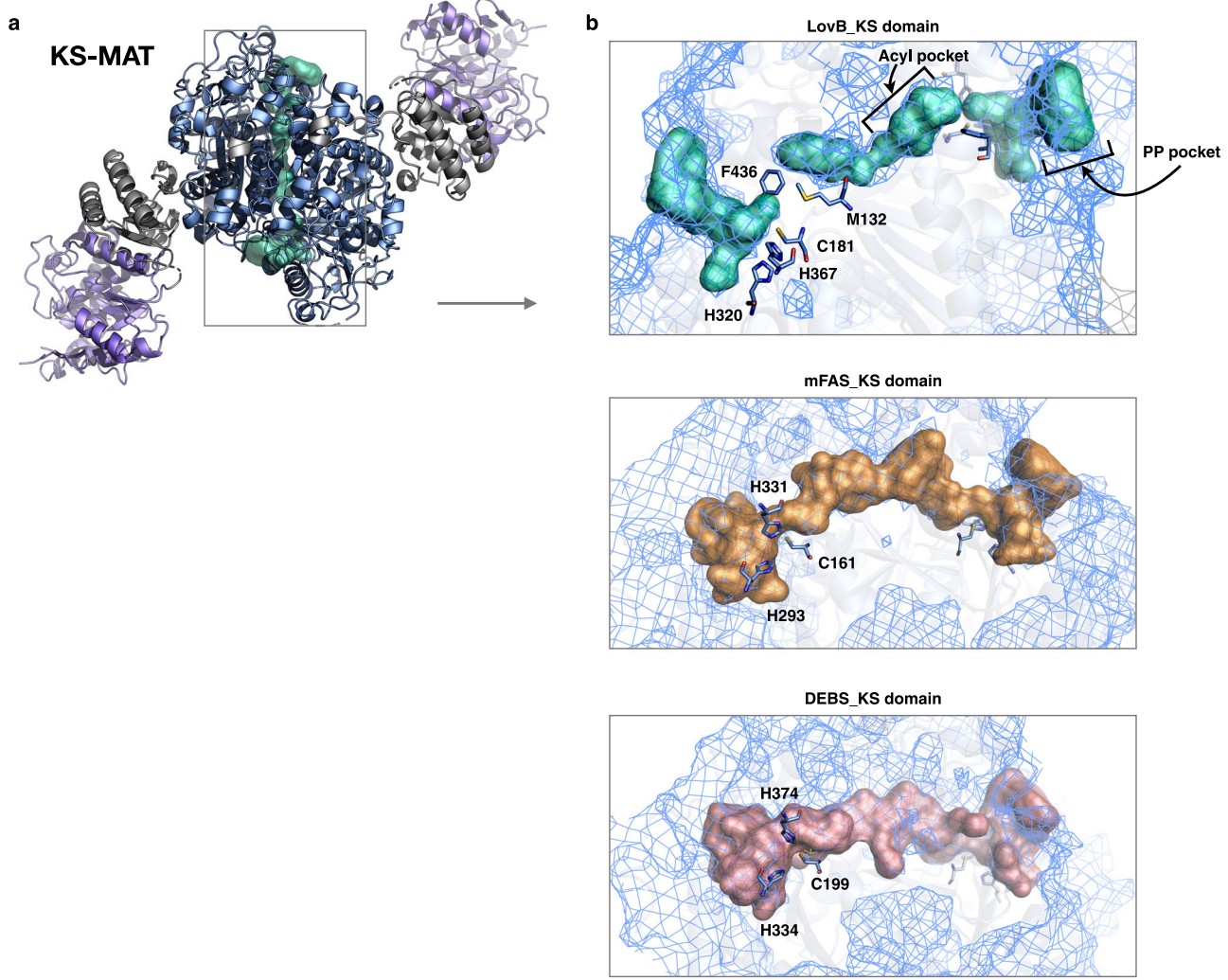

**Fig. 2 The KS–LD–MAT domains of LovB. a** The model of the domains is shown in top view, with the substrate binding tunnel represented in cyan surface. **b** Top, close-up view of the disconnected catalytic tunnel of KS. The conserved active site residues are labeled. Two additional residues (M132 and F436) intercept the acyl (inner) and PPant (outer) pockets. Middle and bottom, the long traversing-through substrate tunnels of KS domains from mFAS and DEBS.

that of mFAS or DEBS KR (Fig. 3b), which might due to that the particles were in the stage that ketoreduction has been completed. A 27 residues length of the post-KR loop was observed to interact with the KR domain with a buried surface of 563 Å². ψKR mainly serve as a structural role for the tailoring region completion, and is unable to bind NADPH due to the truncation of nearly half of the Rossmann fold compared with the fully active KR domain.

The organization of the two DH domains is V-shaped which is similar to mFAS, in contrast to the relatively linear DH domain organization in MAS and modPKSs[15] (Supplementary Fig. 9). Each DH monomer of LovB adopts a pseudodimeric hot-dog fold and harbors the conserved active site residues H985, D1174, and Q1178, which are contributed by both hot-dog folds. The substrate tunnel begins at the α helix near the active site, and has a closed end inside the fold rather than traveling through the entire C-terminal hot dog fold as in mFAS. It is relatively shorter (~11 Å) than the tunnel of mFAS (~18 Å). Six tyrosine and phenylalanine residues surround the tunnel, ready for interaction with the hydrophobic elongated polyketide intermediates in DML synthesis.

The S-adenosyl-methionine (SAM)-dependent CMeT domain comprises two subdomains (Fig. 3d) and resembles the homologous modPKS CurJ (r.m.s deviation of 2.39 Å). The active site residues are located at the two-subdomain interface, and the conserved F1400 represents the substrate entrance region. The binding pocket for SAM is clear, and the location of the hydrophobic substrate cavity between two subdomains is facing towards the catalytic chamber of LovB for the access by ACP, which is necessary for methylation activity during DML synthesis.

The ψER adopts the medium-chain dehydrogenases/reductases MDR fold dimer. It lacks the active site residues, and the substrate tunnel is disrupted, leaving no space for substrate and NADP⁺ cofactor binding (Fig. 3e). This inactive version of the ER dimer mainly contributes to the architecture-fixing role, due to the extensive contacting interface they provide.

**LovB–LovC interaction is essential for DML synthesis**. We observe that 8% of the particles show density for both sides of LovC. Three-dimensional refinements resulted in an overall 4.21 Å resolution map (Supplementary Fig. 4). The EM density allows unambiguous fitting of the LovC crystal structure. However, the resolution at the LovB–LovC interface is too low for the modeling of the protein–protein interaction. 3D classification with a global

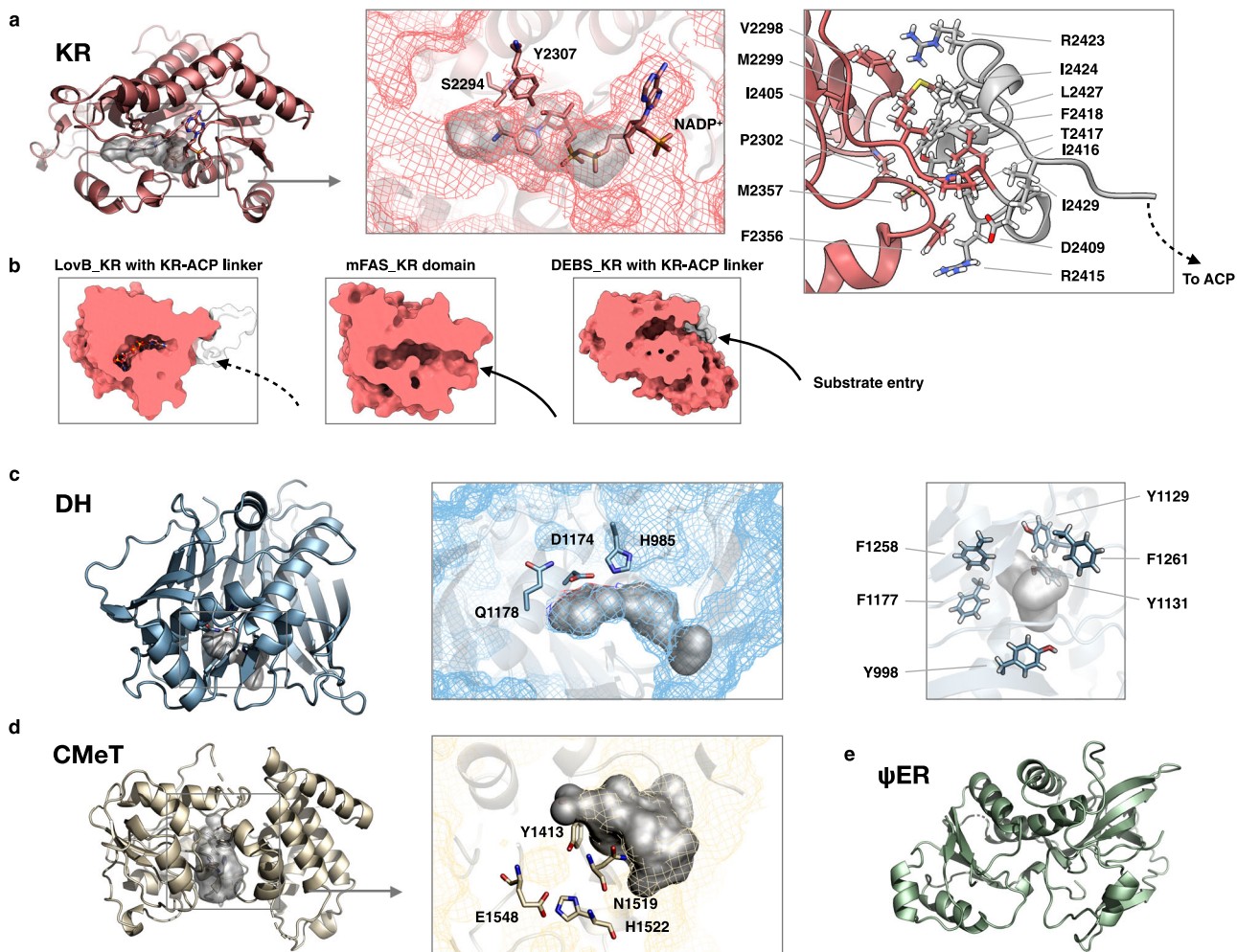

**Fig. 3 Tailoring domains of LovB. a** KR domain with bound cofactor NADP$^+$, substrate tunnel (gray surface) and active site residues. Interacting residues of post-KR linker with KR are labeled. **b** Surface representation of KR with post-KR linker was cut perpendicularly to show the restricted substrate entry groove, compared with the groove for homologous KR domains from mFAS and DEBS. **c** Model of the DH domain. Active site residues are marked, and six aromatic amino acids along the substrate tunnel are highlighted on the right. **d** CMeT domain. Active site residues located between two subdomains are labeled, and the groove formed between the two subdomains is shown in gray surface. **e** ψER domain.

search followed by local finer angular sampling resulted in 40.7% of particles showing a single chain of LovC. The reason that we observed particles with a single LovC chain could be that some of the LovC protein was disordered, or that some LovC fell off during the cryo-EM sample preparation plunge-frozen step. More likely, the particles were not evenly distributed in the solution and had some extent of directional preference (Supplementary Fig. 5d), or LovC was vibrating relatively to the LovB core part. Another possible physiologically relevant cause is that the binding of LovB with LovC has a moderate overall affinity, or there is a negative allosteric crosstalk between them to regulate the synthesis of the polyketide.

To obtain better density integrity for LovC part, the class of map with one LovC chain was further auto-refined and postprocessed. The density of LovC was resolved to the local resolution in the range of 6.1–9.9 Å, which is still not sufficient for precise model placement, but allows the low-resolution guided in silico docking of the LovC crystal structure. To further identify the critical interacting residues and more precisely describe the LovB–C interface, computational docking of the MAT–LovC interacting region (residues 695–757 of MAT) with LovC was performed using RosettaDock[29] (Fig. 4a, b). The 19 lowest-energy models of MAT generated by RosettaCM were used to dock with

LovC, and all structures were kept rigid during the simulation process. The best performance of the first step was used as input for the second round of simulation. Successful docking was based on the formation of an energy funnel with rmsd <2.2 Å from the ten lowest-energy decoys. Figure 4a, b shows that one of the 19 models docks successfully with LovC.

The LovB–C interface is approximately 522 Å$^2$, which is mainly contributed by several residues in the loop region of LovC and α helices of the MAT domain (Fig. 4c). To further verify the binding site, the interaction loops in LovC were mutated (T271L, R272I, K273G, and M274A). In parallel, three mutants of the MAT domain were also designed (MAT Mut1–3). Mut1 had E747A, D748A, E749A, and S750A mutations; Mut2 had H741A and G744A mutations; and Mut3 had D713A and A714S mutations. The mutant of LovC, the MAT domain and its mutants of LovB were cloned, expressed, and purified. Then the purified mutant proteins of the LovC and MAT domain were mixed and incubated for protein complex formation. Figure 4d shows that in contrast to the control incubation that contains the WT LovC and MAT domain protein, which coelute during size exclusion chromatography at 15 ml, mutation of the interacting loop on LovC disrupts the interaction. The LovC mutant and MAT domain protein elute separately. At the same time, the

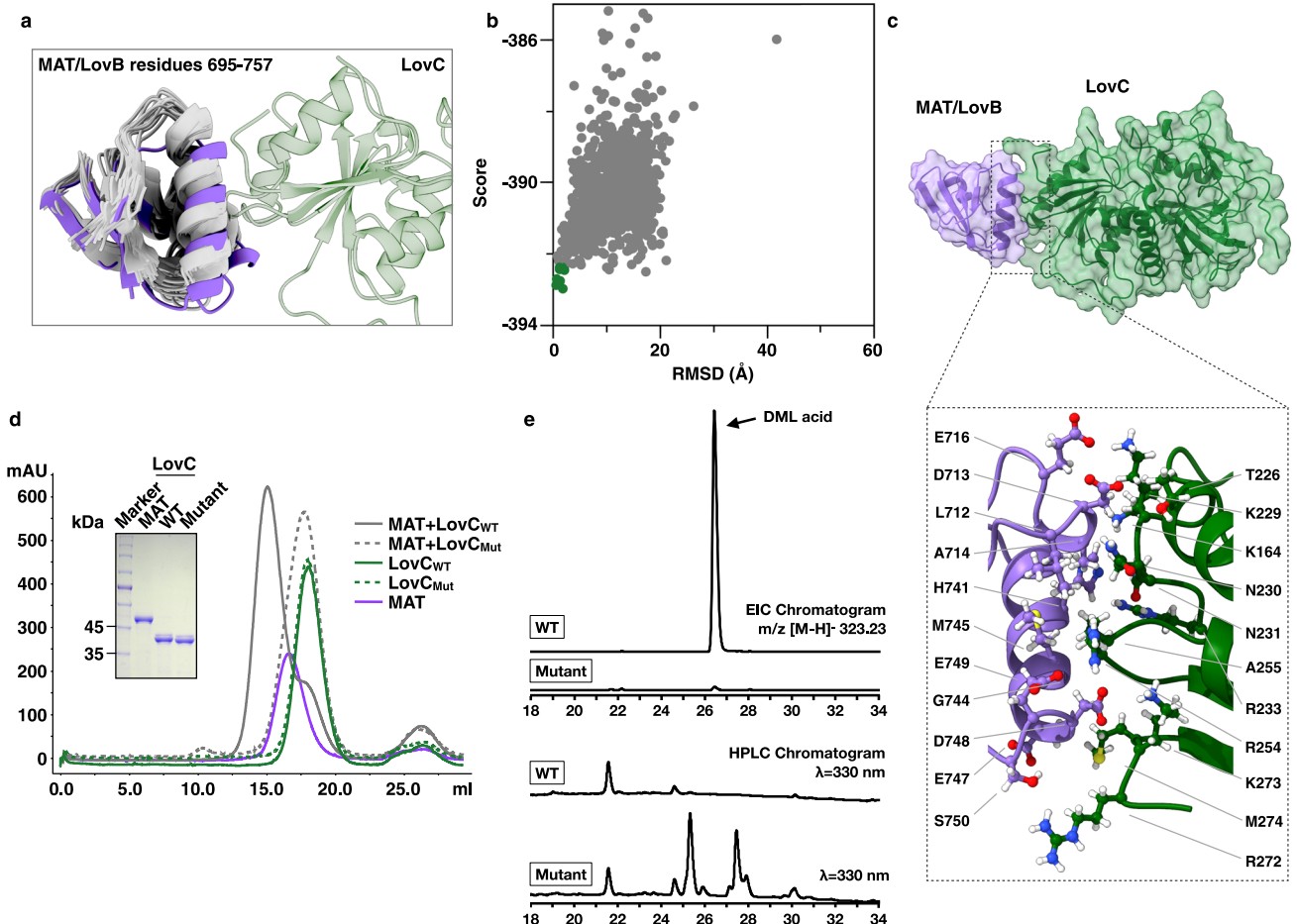

**Fig. 4 Interaction between LovB and LovC is essential for the synthesis of DML. a** Nineteen models built by RosettaCM were chosen for protein docking simulation with LovC. Residues 695–757 (part of the MAT domain) are shown. Purple color denotes the successful docked model. **b** Energy funnel for MAT/LovB–LovC docking analysis. The plot of score vs. rmsd shows the ten lowest-energy decoys (dark green) with rmsd <2.2 Å. **c** LovB–LovC interface. Top, view of the isolated electron density of MAT/LovB and interacting LovC with fitted molecular models. Bottom, close-up view of the interface between MAT/LovB and LovC (amino acids within 4 Å). **d** Size exclusion chromatography profiles for the interaction between LovC and the MAT domain of LovB. The LovC mutant profile is indicated by a dashed line. Elution of the component protein(s) is marked in color. One representative gel panel from at least three independent experiments shows the purified MAT domain of LovB, LovC, and the LovC mutant detected by SDS-PAGE. Source data is provided as a Source Data file. **e** HPLC traces showing the products of the in vitro reactions catalyzed by LovB with LovC or LovC mutant. LovG was included in the reaction mixture to release the final products. The DML acid has calculated and experimentally determined $m/z$ $[M-H]^-$ values of 323.23 and 323.22, respectively. Top, the extracted-ion chromatogram profile; bottom, the HPLC chromatogram profile at $\lambda = 330$ nm.

MAT domain Mut1 abolished interaction with LovC (Supplementary Fig. 10). These observations suggest that the loop within LovC and the helix of MAT domain of LovB are essential for the formation of the LovBC complex.

LovC functions as a gatekeeper for the normal lovastatin synthesis in *A. terreus*[7]. The gate-keeping function is specified by the specific recognition of the intermediates by the active site residues of LovC[19]. The residues interacting with LovB are not involved in the recognition of the substrates. This observation allowed us to test whether the binding of LovC to LovB plays a role, that is, whether the LovBC complex forms an integral catalytic chamber in the catalysis of DML synthesis. The significance of LovB–LovC binding for the integrity of the catalytic chamber was analyzed by an in vitro reaction assay catalyzed by LovB–LovC and LovG. Figure 4e shows that the interface mutation abolished the synthesis of DML acid, in spite of the full catalytic competence of the mutant (Supplementary Fig. 11). The interface mutant synthesized only pyrone shunt products as previously observed for LovB in the absence of LovC (Supplementary Fig. 12). We conclude that the formation of the

LovBC complex is essential for the integrity of the catalytic chamber to the complete total synthesis of DML acid.

## Discussion

HR-iPKSs exist widely in nature, including in animals[30]. The structural basis of the LovB–LovC complex sheds light on the understanding of this family of megasynthases. For the total eight synthetic cycles LovBC uses to produce DML, the polyketide intermediate-tethered ACP domain needs to shuttle back and forth within the catalytic chamber to individual domains (Fig. 5a). The assembly of two LovB monomers between the KS–DH domains shows the interaction between the upper and lower wings (Supplementary Fig. 7), which could sterically hinder the large-scale domain rotation of LovB, which is in contrast with the observations in mFAS[26]. Indeed, dramatic domain rotations in mFAS were not detected in our LovBC samples. Nevertheless, we envisage that the iterative domain interactions between the ACP domain and the catalytic domains are specified by the molecular surface observed here. Further structural study on the

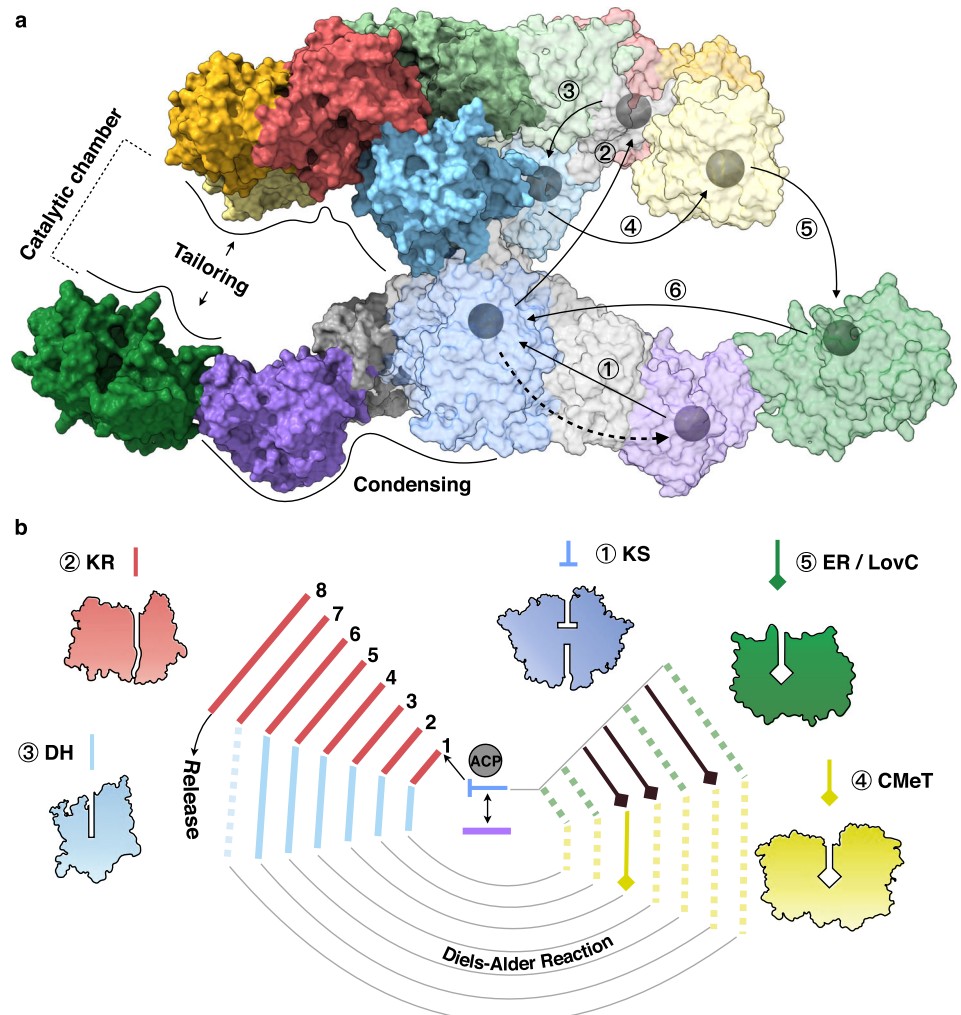

**Fig. 5 Substrate shuttling within the catalytic chamber by the ACP domain. a** The LovBC complex is shown as surface representation. The hypothetical substrate shuttling trajectories (one side) within the catalytic chamber are shown as lines with each arrow pointing towards the next step; the active site residues locations of each domain (depicted in Supplementary Fig. 13) are marked as gray balls. The dashed arrow indicates the loading of the malonyl precursor. **b** Schematic diagrams depicting the iterative, yet permutative function of LovB. After each condensation, the substrate undergoes different tailoring processes. The solid lines indicate the steps in which the intermediates are processed by the domain. The bypassed steps in each cycle are depicted in dashed lines.

interaction of the ever-changing polyketide intermediates with the catalytic tunnels of each domain should reveal, how the HR-iPKS programs the specific permutative functions at each synthetic cycle (Fig. 5b).

## Methods

**Strains, plasmids, and culture conditions**. The strains, plasmids, and primers used in this work are listed in Supplementary Tables 1 and 2. The *Saccharomyces cerevisiae* and *E. coli* strains have been described in the previous publications[7]. YPD medium contains 20 g/l peptone, 10 g/l yeast extract, and 20 g/l dextrose. SC-Uracil dropout medium contains 5 g/l Bacto casamino acids, 6.7 g/l yeast nitrogen base with ammonium sulfate, 20 g/l dextrose, 0.2 g/l adenine hemisulfate, 0.2 g/l tryptophan, and 20 g/l Bacto agar (for solid medium). The Frozen-EZ Yeast Transformation IITMT2001 Kit was purchased from ZYMO RESEARCH CORP. *Saccharomyces cerevisiae* strains were routinely cultured in YPD medium. The yeast plasmid pXW_LovBcH was transformed into *S. cerevisiae* BJ5464-NpgA according to the previously published protocol[7] using the Frozen-EZ Yeast Transformation IITMT2001 Kit.

For the overexpression of LovB, a single colony of *S. cerevisiae* BJ5464-NpgA/pXW_LovBcH was inoculated into 90 ml of SC-Uracil dropout medium in a 250 ml flask and cultured for 48 h at 28 °C with shaking at 220 rpm. Four milliliter aliquots of the culture were then separately inoculated into 1 l of YPD medium in a 3 l flask and cultured for 72 h at 28 °C with shaking at 220 rpm. Cells were harvested by

centrifugation at 4500×*g* for 6 min. Approximately 20 g of cell paste was routinely obtained per 1 l culture. The cell pastes were flash-frozen and stored at −80 °C.

The DNA fragment containing the MAT domain of LovB (55 kDa) was amplified using primers Mat28_18_S and Mat28_18cH_A with plasmid pXW_LovBcH as the template. The vector pET28a was amplified by using primers V28_Mt55_S and V28_Mt55_A. The resulting DNA fragments were fused together using a Trelief™ SoSoo Cloning Kit (Tsingke Biological Technology). Transformation of the fusion product into *E. coli* DH10B generated the expression plasmid pLovB_MATcH. A polyhistidine tag was fused at the C-terminus of the protein.

For the construction of LovB_MAT mutants, pLovB_MATcH was used as a template. The primers were listed in Supplementary Table 2. These resulting PCR products were digested using DpnI to remove the template, and transformed into *E. coli* DH10B. The resulting plasmids were extracted from *E. coli* DH10B. After confirmation by sequencing, the mutational plasmids were transformed into *E. coli* BL21(DE3) for protein expression.

For the construction of the LovC mutant, the plasmid pET28_LovCcH was used as a template, and TRKM to LIGA-S and TRKM to LIGA-A were used as primers. The PCR product was digested using DpnI to remove the template and then transformed into *E. coli* DH10B. The resulting plasmids were isolated from *E. coli* DH10B, confirmed by sequencing and then transformed into *E. coli* BL21(DE3) for protein expression.

Bacterial cells were routinely cultured in Luria broth medium (10 g/l tryptone, 5 g/l yeast extract, and 10 g/l sodium chloride) supplemented with 50 μg/ml kanamycin when needed for strain selection. Specifically, the *E. coli* strains used in protein expression experiments were grown in 1 l of LB medium containing 50 μg/

ml kanamycin at 37 °C with shaking at 220 rpm until the culture optical density at 600 nm ($OD_{600}$) reached 0.6. At this point, gene expression was induced by the addition of isopropyl-D-thiogalactopyranoside (IPTG) to a final concentration of 0.1 mM, and the culture was allowed to incubate for an additional 24 h at 16 °C. Cells were then harvested by centrifugation at 6000×g for 20 min, flash-frozen and stored at −80 °C.

**Purification and sample preparations of His-tagged LovB, LovC and LovG.** Polyethylene glycol-8000 (PEG8000), S-adenosyl-L-methionine (SAM), disuccinimidyl suberate (DSS), dimethylformamide (DMSO), and gravity columns were purchased from Sangon Biotech (Shanghai) Co., Ltd. TWEEN 20 and Millipore's Amicon® Ultra-0.5 10k centrifugal filter devices were purchased from Sigma-Aldrich (Merck KGaA, Darmstadt, Germany). Ni-NTA resin and Superose 6 Increase 10/300 GL columns were purchased from GE Healthcare (GE Healthcare Life Sciences, Little Chalfont, UK). All experiments were performed at 4 °C unless indicated.

For the purification of LovC, 5 g of frozen cells were thawed, resuspended in 50 ml of buffer A (50 mM Tris·HCl pH 8.0, 150 mM NaCl, 5% glycerol, 40 mM imidazole) and lysed using a French press (Union-Biotech, Shanghai, China) operated at 4 °C. Cell debris was removed by centrifugation at 18,000×g for 30 min, and the resulting supernatant was loaded onto a 12 ml gravity-flow column packed with 2 ml of Ni-NTA resin pre-equilibrated with 20 ml of buffer A. The resin was then washed with 40 ml of buffer A. LovC was eluted using 5 ml of elution buffer (300 mM imidazole pH 8.0, 50 mM NaCl, 5% glycerol, 4 mM SAM). LovC was incubated on ice until its usage in the LovBC complex formation. The concentration of purified LovC was measured using the Bradford assay. Approximately 3 mg of LovC can routinely be obtained from 5 g of cell paste. The fluorometric activity assays of LovC and the mutants were carried out according to Ames et al.[19]. Purification of LovG was according to Xu et al[3]. Briefly, 3 g of frozen cells were thawed, resuspended in 30 ml of buffer A and lysed using the French press operated at 4 °C. Cell debris was removed by centrifugation at 18,000×g for 30 min, and the resulting supernatant was loaded onto a 12 ml gravity-flow column packed with 2 ml of Ni-NTA resin pre-equilibrated with 20 ml of buffer A. The resin was then washed with 40 mL of buffer A. LovG was eluted using 5 ml of elution buffer. Approximately 3 mg of LovG can routinely be obtained from 3 g of cell paste.

For the purification of LovB, 50 g of frozen cells were thawed, resuspended in 100 ml of buffer A and lysed by a French press operated at 4 °C (1100 bar). Cell debris was removed by centrifugation at 18,000×g for 60 min, and the resulting supernatant was precipitated using PEG8000. PEG8000 stock solution (50% w/v dissolved in 100 mM Tris pH 8.0) was added to the supernatant drop by drop slowly with stirring until the final concentration reached 8%. The solution was stirred for an additional 30 min and then centrifuged at 16,000×g for 10 min. The supernatant was discarded, and the resulting pellet was dissolved in 50 ml of buffer A. After centrifugation at 16,000×g and 4 °C for 10 min, the supernatant was aliquoted into two sterile 50 ml conical tubes, each containing 3 ml of Ni-NTA resin pre-equilibrated with buffer A and then gently rotated for 2 h using a QB-206 multipurpose shaker (Kylin-Bell, Haimen, China) for sufficient protein–resin interaction. After spinning at 800 g for 3 min, the supernatant was discarded, and a total of 6 ml of resin was transferred to a 20 ml gravity column using buffer A. The column was then washed with 60 ml of buffer A and eluted with 20 ml of elution buffer. The eluted liquid was collected in 5 ml aliquots. The two aliquots with the highest concentration were combined. Approximately 6 mg of LovB protein in 8 ml can be obtained from 50 g of frozen cells.

Six milligrams of LovB was mixed with 1 mg of LovC and 2 mM of each of the three cofactors (Mal-CoA, NADPH, and SAM) and incubated for at least 1 h. Then, 300 µl of 100 mM DSS (dissolved in DMSO) crosslinker was added to the mixture. The mixture was incubated on ice for 2 h for efficient crosslinking. The crosslinking reaction was quenched by adding 1 M Tris (pH 8.0) stock solution to a final concentration of 50 mM and incubating for an additional 30 min. The crosslinked LovBC solution was concentrated to 1 ml using a Millipore's Amicon® Ultra 10k centrifugal filter device according to the protocol provided by the company. The solution was centrifuged at 16,000×g for 10 min to remove precipitates and then subjected to size exclusion chromatography using a pre-equilibrated Superose 6 Increase 10/300 GL column in sizing buffer (50 mM Tricine pH 8.0, 4 mM SAM) on an ÄKTA fast protein liquid chromatography system (GE Healthcare Life Sciences). The peak fractions were pooled and concentrated with the centrifugal filter device to a concentration of approximately 8 mg/ml (determined by Bradford assay using BSA as a standard). The sample was then added to a final concentration of 2 mM each of the three cofactors (Mal-CoA, NADPH, and SAM) again for cryo-EM specimen preparation.

**Detection of DML acid from in vitro reconstitution experiments.** Twenty-five micromolar of LovB was incubated with 25 µM WT or mutant of LovC, 25 µM LovG, 2 mM Malonyl-CoA, 2 mM NADPH, and 2 mM SAM in buffer (100 mM $NaH_2PO_4$, pH 7.4, 10% glycerol, 2 mM DTT, 2 mM EDTA) in a 250 µl solution at 25 °C for 24 h. Reactions were quenched and extracted twice with an equal volume of 99% ethyl acetate (EA)/1% acetic acid (AcOH). The organic phase was evaporated to dryness, and redissolved in 0.05 M NaOH in 15 µl of methanol and analyzed by LC-MS. LC-MS was conducted with an Agilent 1290 Infinity Liquid chromatography and 6545 Quadrupole Time-of-Flight Mass Spectrometer by using

negative electrospray ionization and an Agilent 5µ 4.6 × 150 mm C18 reverse-phase column. Samples were separated at room temperature on a linear gradient of 5–95% $CH_3CN$ (v/v) in $H_2O$ supplemented with 0.05% (v/v) formic acid over 30 min, and held at 95% $CH_3CN$/ 0.05% formic acid for 30 min at a flow rate of 0.4 ml/min.

**Fluorometric assay.** The fluorometric activity assay was carried out using a Bio-Tek Synergy 2 Multi-mode Microplate Reader with EX set to 340 nm and the EM set to 455 nm to follow the disappearance of $EM_{455}$, as NADPH was oxidized in the presence of substrate over time. Twenty-five micromolar of LovC (WT or Interface mutant or Active site residues mutant) was preincubated with 100 µM NADPH and added to the reaction solution (100 mM $KH_2PO_4$, pH 7.0, 2% DMSO, 200 µM Crotonoyl-CoA) in a total of 100 µl volumn in a Greiner 96 flat bottom plate. After quickly mixing the solution by pipetting, a total of 10 min scan with 20 s intervals was performed monitoring $EM_{455}$ at 25 °C. The mean value of relative fluorescence units (RFU) difference from 0 (Max) to 10 (Min) minutes was calculated. The relative activity of LovC Interface mutant was determined by the RFU difference value ratio to WT, which serve as positive control, and LovC active site residues mutant as negative control.

**Cryo-EM specimen preparation and data acquisition.** Immediately prior to specimen preparation, Tween 20 (10%) was added to the freshly purified LovBC complex to a final concentration of 0.1%, which improved the quality of vitreous ice in the specimen. Four microliter aliquots of specimen at ~8 mg/ml were applied to glow-discharged holey carbon grids (Quantifoil Cu, R1.2/1.3, 200 mesh) for 60 s of incubation and then blotted for 2.5 s and plunge-frozen into liquid ethane precooled by liquid nitrogen using a Vitrobot Mark IV (FEI) operated at approximately 100% humidity and 22 °C. Cryo-EM images were collected with a Titan Krios electron microscope (FEI) operated at 300 kV and equipped with a K2 Summit direct electron detector (Gatan). Thirty-eight frames were recorded for each movie stack at a nominal magnification of 22,500-fold in super resolution mode with a pixel size of 1.0 Å in the defocus range of 1.5–2.5 µm. A total of 8136 movie stacks of the LovBC complex were automatically collected using SerialEM[20] with an exposure time of 7.6 s (0.2 s per frame) and a total dose of 60.8 e−/Å$^2$.

**Image processing.** All movies of datasets were aligned and dose-weighted using MotionCor2[21]. Contrast transfer function (CTF) and defocus parameters were determined by Gctf[22]. Micrograph checking, particle autopicking, 2D, 3D classification, autorefinement, postprocessing, and resolution estimation of each density map were performed using RELION 3.0[23–25]. Approximately 2000 particles of each dataset were manually picked and subjected to reference-free 2D classification. The best representative 2D classes were selected as templates for autopicking.

For the reconstruction of the LovBC complex, datasets were cleaned by removing ice contaminants and junk particles after two rounds of 2D classification, and good classes were kept to generate the 30 Å 3D initial model, which was low-pass filtered to 70 Å as the reference for subsequent 3D classification (Supplementary Fig. 4). One class of 16,444 particles was auto-refined and postprocessed, yielding the reconstruction of a 4.21 Å LovBC complex. The best classes of 205,047 particles were selected with a soft mask on the core (LovB) for focused 3D auto-refinement, postprocessing, CTF refinement and Bayesian polishing, yielding the reconstruction of a 3.09 Å LovB density map. 3D classification with finer, local angular searches was further performed for conformational difference detection. For better integrity of the LovC part of the LovBC complex, a total of 83,573 particles were auto-refined and postprocessed, yielding the reconstruction of a 3.60 Å LovB$_C$ (with one side of LovC only) density map. For better resolution of the lovB, one class of 75,036 particles was imposed in parallel with C1 and C2 symmetry and auto-refined, yielding the density maps of 3.53 Å and 3.27 Å, respectively. No differences between the C1 and C2 maps were detected when inspecting in Coot. Finally, classes were combined and imposed with C2 symmetry. A total of 205,047 particles were further processed by focused 3D auto-refinement using a soft mask around the LovB with signal subtraction, yielding the reconstruction of the 2.91 Å LovB density map (Supplementary Fig. 5). The resolution of all density maps was estimated based on the corrected gold standard Fourier shell correlation (FSC) at the 0.143 criterion.

**Model building and refinement.** First, the HHpred server was used for protein homology analysis using the HMM–HMM comparison method[31,32]. Multiple homologous crystal structures for each domain of LovB (KS, MAT, DH, CMeT, ψKR, ψER, and KR) were rigid-body fitted into the density using UCSF-Chimera[33] for comparative model rebuilding using RosettaCM[34–36]. The resulting atomic coordinates were further manually adjusted and built using Coot[37] and ISOLDE[38]. Structure refinement was carried out by Phenix in real space with secondary structure and geometry restraints to prevent over-fitting[39]. Finally, MolProbity[40] was used for model validation. The statistics are summarized in Supplementary Table 3. All cryo-EM densities and atomic models were visualized, and the figures depicting them were prepared in PyMOL, UCSF-Chimera, and ChimeraX[41].

**Substrate tunnel generation and protein docking.** Substrate tunnels within each domain of lovB were calculated with the program Hollow[42] and adjusted in PyMOL (The PyMOL Molecular Graphics System, Version 2.0 Schrödinger, LLC.).

Docking between MAT/LovB (residues 695–757) and LovC was performed using RosettaDock in Rosetta v3.2[29] using the online Rosie server[43,44]. Nineteen out of the 1000 lowest-energy MAT/LovB models generated by the first round of RosettaCM were selected for docking simulation with LovC (both fitted to the experimental density). The lowest-energy docking run decoy of one best simulation (out of 19) was used as the input for the second round of docking. The ten lowest-energy decoys with rmsd < 2.2 Å indicate docking success.

**Reporting summary.** Further information on research design is available in the Nature Research Reporting Summary linked to this article.

## Data availability

The 3D cryo-EM density maps in this study have been deposited in the Electron Microscopy Data Bank (https://www.emdataresource.org/) with accession numbers EMD-30434 and EMD-30435 (Supplementary Table 3). The atomic coordinates have been deposited in the Protein Data Bank as 7CPX and 7CPY. The atomic coordinate of LovB–C interface underlying Fig. 4c is provided as a Supplementary Data 1. Other data are available from the corresponding authors upon request. Source data are provided with this paper.

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

## Acknowledgements

We thank Z. Liu, F. F. Wang, G. Y. Li, L. H. Xin, J. L. Duan, and N. Liu for their help in the sample preparation and data collection. Cryo-EM images were collected at the National Facility for Protein Science in Shanghai (NFPS), Zhangjiang Lab. The computations in this paper were run on the π 2.0 cluster supported by the Center for High Performance Computing at Shanghai Jiao Tong University. This work was financially supported by National Key R&D Program of China (2018YFA0900700, 2019YFA0905400), the Ministry of Science and Technology (2015CB554203), the National Science Foundation of China (91753123, 31470830, 21661140002).

## Author contributions

Z.W., Y.T., and J.W. conceived the study. J.W., J.L., C.L. L.K., C.P., and C.S. performed the experiments. Z.W., J.W., Y.T. J.L. W.Z., and L.K. analyzed the data. All authors wrote the paper. Z.W., Z.D., and J.L. supervised this project.

## Competing interests

The authors declare no competing interests.
