## [Peer Review File · Nature Communications]

REVIEWER COMMENTS

Reviewer #1 (Remarks to the Author):

Understanding the complex mechanisms of polyketide synthases (PKSs) is highly relevant and a crucial step towards engineering the production of polyketides, a highly important compound class in drug discovery. Wang et al. present the cryoEM structure of a highly-reducing iterative (PKS) involved in lovastatin biosynthesis, LovB, in isolation and in complex with its partner protein LovC, an trans-acting enoyl reductase. The core part of LovB, from N-terminus to the flexibly tethered acyl carrier protein, is closely related in domain organization to the iterative fatty acid synthase (FAS), but distinguished by its programmed selective use of domains in individual precursor elongation cycles. The structural data for this core part of LovB are of excellent quality and reveal an unexpectedly close relation to FAS at the overall structural level. However, the flexibly tethered ACP and following C domain remain fully disordered. The previously unknown binding site of LovC on LovB was revealed by cryoEM, but apparently the resolution of the LovC region was so low, that in silico docking was required to obtain a model of the complex. The authors aim to support their model by providing activity data on an interface mutant of LovC and further describe the individual active sites of the resolved catalytic domains. Unfortunately, this analysis remains rather descriptive and isn't suggesting specific mechanisms for the unique programming mechanism of polyketide biosynthesis by LovB.

For detailed comments, see below

I.21: Add "resolution"

I.23: better use: ...allows the completion of a ...

I.53: "Megasyntase programming has remained a mystery": While this is true for exactly LovB, it might be worth recognizing earlier work that provided substantial insights into programmed biosynthesis of iterative PKS in the past few years, e.g. by the Cox and Townsend labs.

I.101: Subheading not clearly comprehensible; do you mean: "Structural analysis of LovB enzymatic domains"

I.101 following: The protein complex was incubated during preparation with Mal-CoA, NADPH and SAM, but none of the cofactors are resolved in the cryoEM analysis. I assume they were removed in SEC analysis: Could the authors confirm and explicitly mention this in the text ? Is there any reason why cofactors weren't re-added ?

I.140: Note that "MT" is used in the text but "MET" in the corresponding figure.

I.151: following: Report in the text the local resolution of lovC, as well as the preferential observation in one protomer. Clearly state in the main text, whether density alone is sufficient for model placement or whether the LovB-LovC interaction is described by a low-res density guided in silico docking.

I.157: Please briefly describe the mutations in the text.

I.158: What does "incubated for protein-protein interaction" mean ?

I.162: An important missing control is to quantitatively demonstrate full catalytic competence of the mutant LovC variants.

I.182ff: This statement is very generic, " the detailed constitution (sic!) in the catalytic tunnelcould exert their selectivity ..." and doesn't provide novel insights, thus the following sentence might be a bit of an overstatement. It would definitely increase the impact if the authors could analyze substrate pocket to actually provide mechanisms or at least testable hypothesis on mechanisms.

Fig. 2: Indicate for less experienced users, which part of the tunnel has which role ? If the inner part is disconnected from the outer part, does it still play a role in substrate binding ?

Fig. 4: Use of modelling and docking and its necessity for structural analysis should be discussed also in the text. Also the assay introduced here deserved at least an extended mentioning in the text.

Line 445: "(with one side of LovC only)": This should be mentioned and importantly be discussed in the text.

Extended data Table 3: Model Compositions: EMD-30435 should have 3 or 4 chains ?

Reviewer #2 (Remarks to the Author):

Statins are inhibitors of an enzyme involved in cholesterol biosynthesis and are used to treat hypercholesterolemia. The iterative polyketide synthase lovastatin nonaketide synthase (LovB) together with the trans-acting enoyl reductase LovC synthesizes an intermediate in the biosynthesis of lovastatin. Wang et al. have produced single-particle cryo-EM maps of LovB and of its complex with LovC. The structure turned out to be surprisingly rigid compared to the related, well-studied mammalian fatty acid synthase, and a tracing of the complete LovB was possible with the exception of the acyl carrier protein and the C-terminal domain following it. Although LovB and LovC were mixed before freezing, only a minority of the particles showed density for LovC, and only at one of the monomers in the dimer. Nevertheless, the structure of the complex enabled the docking of a crystal structure and the interface was verified by mutation analysis.

This is a carefully executed cryo-EM study that yields new insights in the structure of megasynthases. It will be of high interest to experts in the field. The paper is overall well-written in terms of scientific content; it would however benefit strongly from editing by a native English speaker.

A few points have to be clarified to make the paper suitable for publication.

1. Abstract: line 15-17 states that LovB builds lovastatin from DML. However, DML is the product of the synthesis.
2. Line 23: "the fully formation" is unclear.
3. Line 25: "structural specific way" is unclear.
4. Line 132-133: "the relative linear DH dimers of MAS" is unclear.

5. Line 176-178: this sentence is completely unclear and needs rewriting.
6. The authors consistently use the term “active sites” where they mean “active site residues”. This should be corrected.
7. Line 426: $60.8 \text{ e}/\text{\AA}^2$ is the total dose, not the dose rate.
8. Figure 1e: the lettering of ACP and also LD is too small.
9. Figure 3d: The MT domain is labelled “MET” on the figure.
10. Figure 4a: it is unclear where LovB and LovC are in this figure.
11. Extended Data Figure 5: this figure is very uninformative. To demonstrate the quality of the 2.9 Å resolution cryo-EM map, please show smaller regions with side chains.
12. Extended Data Table 3: The number of atoms and residues is listed as the same for both structures, but one is labeled LovB and the other LovB with LovC. Either the numbers or the labels must be wrong.
13. Extended Data Table 3: the pixel size is listed as 1 Å, this needs more digits (1.00).

Point-by-point response to the reviewers

*Reviewer Comments: Black, Helvetica, 9

*Author Responses: Blue, Times New Roman, 10.5

REVIEWER COMMENTS

Reviewer #1 (Remarks to the Author):

Understanding the complex mechanisms of polyketide synthases (PKSs) is highly relevant and a crucial step towards engineering the production of polyketides, a highly important compound class in drug discovery. Wang et al. present the cryoEM structure of a highly-reducing iterative (PKS) involved in lovastatin biosynthesis, LovB, in isolation and in complex with its partner protein LovC, an trans-acting enoyl reductase. The core part of LovB, from N-terminus to the flexibly tethered acyl carrier protein, is closely related in domain organization to the iterative fatty acid synthase (FAS), but distinguished by its programmed selective use of domains in individual precursor elongation cycles. The structural data for this core part of LovB are of excellent quality and reveal an unexpectedly close relation to FAS at the overall structural level. However, the flexibly tethered ACP and following C domain remain fully disordered. The previously unknown binding site of LovC on LovB was revealed by cryoEM, but apparently the resolution of the LovC region was so low, that in silico docking was required to obtain a model of the complex. The authors aim to support their model by providing activity data on an interface mutant of LovC and further describe the individual active sites of the resolved catalytic domains. Unfortunately, this analysis remains rather descriptive and isn't suggesting specific mechanisms for the unique programming mechanism of polyketide biosynthesis by LovB.

We are thankful for the reviewer's thorough and critical comments on our work. These comments are very helpful to the quality promotion of this manuscript. The points have been addressed point by point, and new experiments have been performed for better supporting our conclusion.

For detailed comments, see below

I.21: Add "resolution"

A: Thanks. Line 21 has been edited as: "Here, we present the cryo-EM structure of the LovB-LovC complex at 3.60 Å and the core LovB at 2.91 Å resolution."

I.23: better use: ...allows the completion of a ...

A: Line 23 has been revised as: "The binding of LovC laterally to the malonyl-acetyl transferase (MAT) domain allows the completion of a "L"-shaped catalytic chamber consisting of six active domains."

I.53: "Megasyntase programming has remained a mystery": While this is true for exactly LovB, it might be worth recognizing earlier work that provided substantial insights into programmed biosynthesis of iterative PKS in the past few years, e.g. by the Cox and Townsend labs.

A: Thanks. We have carefully read works done by the Cox and Townsend labs which provide important insights into the programming of iterative PKS. The main text in the introduction section was then revised with the papers cited.

Line 55: "Excellent domain swapping experimental results and biochemical assays of an isolated enoyl reductase from the Cox and Townsend labs have demonstrated that the individual catalytic modifying domains themselves possess selectivity for specific substrates^{9,10}. The CMeT domain, which catalyzes methylation in polyketide formation for citrinin, functions as a gatekeeper¹¹. The starter unit selection carried out by the SAT domain for PksA also contribute to the selectivity¹². These observations led to the proposal that in the programming of iterative PKSs, ultimate control resides in the structure of the protein and the recognition of structurally ever-changing substrates⁹."

9. Fisch, K. M. *et al.* Rational domain swaps decipher programming in fungal highly reducing polyketide synthases and resurrect an extinct metabolite. *Journal of the American Chemical Society* **133**, 16635–16641 (2011).

10. Roberts, D. M. *et al.* Substrate selectivity of an isolated enoyl reductase catalytic domain from an iterative highly reducing fungal polyketide synthase reveals key components of programming. *Chem Sci* **8**, 1116–1126 (2017).
11. Storm, P. A., Herbst, D. A., Maier, T. & Townsend, C. A. Functional and Structural Analysis of Programmed C-Methylation in the Biosynthesis of the Fungal Polyketide Citrinin. *Cell Chem Biol* **24**, 316–325 (2017).
12. Foulke-Abel, J. & Townsend, C. A. Demonstration of starter unit interprotein transfer from a fatty acid synthase to a multidomain, nonreducing polyketide synthase. *ChemBioChem* **13**, 1880–1884 (2012).

I.101: Subheading not clearly comprehensible; do you mean: “Structural analysis of LovB enzymatic domains”

A: The subheading is revised to “Structural analysis of LovB enzymatic domains” in Line 113.

I.101 following: The protein complex was incubated during preparation with Mal-CoA, NADPH and SAM, but none of the cofactors are resolved in the cryoEM analysis. I assume they were removed in SEC analysis: Could the authors confirm and explicitly mention this in the text ? Is there any reason why cofactors weren't re-added ?

A: The cofactors were re-added after the SEC purification step.

Actually, when we optimized the conditions for LovBC cryo-EM sample preparation using 200 kV EM, the 2D classification results show that when the cofactors were removed after the SEC step, the LovBC particles were heterogeneous and tended to aggregate, compared with the particles obtained when the cofactors were re-added after the SEC step. None of the cofactors were observed in our cryo-EM maps, but we don't know why. It could possibly be due to that the majority of the protein complex were pushed to the conformation representing the final stage of the DML synthesis just before releasing.

Line 80 revised: “After SEC analysis, three cofactors were re-added into the LovBC solution for cryo-EM sample preparation, ensuring that the particles were homogeneous enough for structure determination (Supplementary Fig. 3).” Supplementary Fig. 3 has been added to the supplementary information.

LovBC_added with cofactors

LovBC_no cofactors

Supplementary Fig. 3 LovBC cryo-EM sample optimization. Representative micrographs and 2D classifications of protein particles prepared in the presence or absence of the cofactors (2 mM NADPH, 2 mM SAM, and 2 mM Malonyl-CoA).

I.140: Note that “MT” is used in the text but “MET” in the corresponding figure.

A: Thanks. “CMeT” is used in Russell J. Cox and Craig A. Townsend *et al.*'s works, which is better. We have now corrected the “MT” to “CMeT” in Fig. 3d and throughout the text.

d

I.151: following: Report in the text the local resolution of lovC, as well as the preferential observation in one protomer. Clearly state in the main text, whether density alone is sufficient for model placement or whether the LovB-LovC interaction is described by a low-res density guided in silico docking.

A: Line 179 revised as: “The density of LovC was resolved to the local resolution in the range of 6.1-9.9 Å, which is still not sufficient for precise model placement, but allows the low-resolution guided in silico docking of the LovC crystal structure.”

Please see also our answer on the question of “(with one side of LovC only)”.

I.157: Please briefly describe the mutations in the text.

A: As suggested, we have now described the three mutants of MAT domain in Line 192: “In parallel, three mutants of the MAT domain were also designed (MAT Mut1-3). Mut1 had E747A, D748A, E749A, and S750A mutations; Mut2 had H741A and G744A mutations; and Mut3 had D713A and A714S mutations.”

I.158: What does “incubated for protein-protein interaction” mean ?

A: We incubated LovC with MAT (WT/Mut) domain protein for protein complex formation.

We have revised this sentence in Line 194: “The mutant of LovC, the MAT domain and its mutants of LovB were cloned, expressed, and purified. Then the purified mutant proteins of the LovC and MAT domain were mixed and incubated for protein complex formation.”

I.162: An important missing control is to quantitatively demonstrate full catalytic competence of the mutant LovC variants.

A: As suggested, we have assayed the activity of the interface mutant LovC variant by monitoring the relative fluorescence units (RFU) of NADPH, and compared with LovC WT and the active site residues mutant. Supplementary Fig. 11 has been added to the supplementary information.

Line 209: “Fig. 4e shows that the interface mutation abolished the synthesis of DML acid, in spite of the full catalytic competence of the mutant (Supplementary Fig. 11).”

Supplementary Fig. 11 Activity assay of LovC interface mutant. The plot shows relative activities of LovC WT (-●-), the interface mutant (-●-) and the active site residues mutant (-●-).

I.182ff: This statement is very generic, “the detailed constitution (sic!) in the catalytic tunnelcould exert their selectivity ...” and doesn’t provide novel insights, thus the following sentence might be a bit of an overstatement. It would definitely increase the impact if the authors could analyze substrate pocket to actually provide mechanisms or at least testable hypothesis on mechanisms.

A: The missing of the structural information on the substrate intermediates-catalytic residues interaction hampers a solid mechanism proposal. The last sentence is deleted and Line 226 is revised to: “Further structural study on the interaction of the ever-changing polyketide intermediates with the catalytic tunnels of each domain should reveal how the HR-iPKS programs the specific permutative functions at each synthetic cycle (Fig. 5b).”

Fig. 2: Indicate for less experienced users, which part of the tunnel has which role? If the inner part is disconnected from the outer part, does it still play a role in substrate binding ?

A: We have added the label (“PP pocket and acyl pocket”) into Fig. 2b, and briefly discussed the roles in the Line 131 for distinguishing between the inner (acyl pocket) part and outer (PP pocket) part. Two references describing the function of “gates” have been cited.

Line 131: “Although the acyl pocket tunnel is disconnected from the PP pocket tunnel observed here, it might represent a conformational state in the absence of an intermediate in the tunnel. The hydrophobic interacting residue pair (F436 and M132) could potentially function as part of a gate^{25,26}, that dynamically controls substrate access to the active site, prevents solvent access to specific regions of the protein, or synchronizes processes occurring in distant parts of the megasynthase.”

25. Kingsley, L. J. & Lill, M. A. Substrate tunnels in enzymes: structure-function relationships and computational methodology. *Proteins* **83**, 599–611 (2015).
26. Gora, A., Brezovsky, J. & Damborsky, J. Gates of enzymes. *Chem Rev* **113**, 5871–5923 (2013).

Fig. 4: Use of modelling and docking and its necessity for structural analysis should be discussed also in the text. Line 445: “(with one side of LovC only)”: This should be mentioned and importantly be discussed in the text.

A: We have now discussed the observation of one side LovC, the necessity and process for structural docking analysis in Line 168: “We observe that 8% of the particles show density for both sides of LovC. Three-dimensional refinements resulted in an overall 4.21 Å resolution map (Supplementary Fig. 4). The EM density allows the unambiguous fitting of the LovC crystal structure. However, the resolution at the LovB-LovC interface is too low for the modeling of the protein-protein interaction. 3D classification with a global search followed by local finer angular sampling resulted in 40.7% of particles showing a single chain of LovC. The reasons that we observed particles with a single LovC chain could be that some of the LovC protein was disordered, or that some LovC felled off during the cryo-EM sample preparation plunge-frozen step. More likely, the particles were not evenly distributed in the solution and had some extent of directional preference (Supplementary Fig. 5d), or LovC was vibrating relatively to the LovB core part.

To obtain better density integrity of LovC part, the class of map with one LovC chain was further auto-refined and postprocessed. The density of LovC was resolved to the local resolution in the range of 6.1-9.9 Å, which is still not sufficient for precise model placement, but allows the low-resolution guided in silico docking of the LovC crystal structure. To further identify the critical interacting residues and more precisely describe the LovB-C interface, computational docking of the MAT-LovC interacting region (residues 695 to 757 of MAT) with LovC was performed using RosettaDock²⁷ (Fig. 4a-b).”

Also the assay introduced here deserved at least an extended mentioning in the text.

A: We revised the text, with a brief introduction of why we assayed the activity of the LovC mutants. Line 203: “LovC functions as a gatekeeper for the normal lovastatin synthesis in *A. terreus*⁷. The gate-keeping function is specified by the specific recognition of the intermediates by the active site residues of LovC¹⁹. The residues interacting with LovB are not involved in the recognition of the substrates. This

observation allowed us to test whether the binding of LovC to LovB plays a role, that is, whether the LovBC complex forms an integral catalytic chamber in the catalysis of DML synthesis.”

Extended data Table 3: Model Compositions: EMD-30435 should have 3 or 4 chains ?

A: We have now corrected the chain number to 4 for EMD-30435 (Supplementary Table 3), with two chains of LovB plus two chains of LovC.

Reviewer #2 (Remarks to the Author):

Statins are inhibitors of an enzyme involved in cholesterol biosynthesis and are used to treat hypercholesterolemia. The iterative polyketide synthase lovastatin nonaketide synthase (LovB) together with the trans-acting enoyl reductase LovC synthesizes an intermediate in the biosynthesis of lovastatin. Wang et al. have produced single-particle cryo-EM maps of LovB and of its complex with LovC. The structure turned out to be surprisingly rigid compared to the related, well-studied mammalian fatty acid synthase, and a tracing of the complete LovB was possible with the exception of the acyl carrier protein and the C-terminal domain following it. Although LovB and LovC were mixed before freezing, only a minority of the particles showed density for LovC, and only at one of the monomers in the dimer. Nevertheless, the structure of the complex enabled the docking of a crystal structure and the interface was verified by mutation analysis.

This is a carefully executed cryo-EM study that yields new insights in the structure of megasynthases. It will be of high interest to experts in the field. The paper is overall well-written in terms of scientific content; it would however benefit strongly from editing by a native English speaker.

We are grateful for the reviewer’s comments on our work. The points have been addressed point by point. The manuscript has also been sent for language polishing by Springer Nature Company. Please see Supplementary Information Springer Nature.

A few points have to be clarified to make the paper suitable for publication.

1. Abstract: line 15-17 states that LovB builds lovastatin from DML. However, DML is the product of the synthesis.

A: Thanks. We have revised this sentence in Line 16: “Lovastatin, one of the precursors of statins, is formed from dihydromonacolin L (DML), which is synthesized by lovastatin nonaketide synthase (LovB), with the assistance of a separate *trans*-acting enoyl reductase (LovC)¹⁻⁵”

2. Line 23: “the fully formation” is unclear.

A: For more clarity, we have changed this sentence to Line 23: “The binding of LovC laterally to the malonyl-acetyl transferase (MAT) domain allows the completion of a “L”-shaped catalytic chamber consisting of six active domains.”

3. Line 25: “structural specific way” is unclear.

A: Line 25 revised: “This architecture and the structural details of the megasynthase provide the basis for the processing of the intermediates by the individual catalytic domains.”

4. Line 132-133: “the relative linear DH dimers of MAS” is unclear.

A: We have now added a new figure as Supplementary Fig. 9 for better illustrating the DH dimers difference between LovB/mFAS and MAS/CurJ.

Line 132-133 revised in Line 149: “The organization of the two DH domains is “V”-shaped which is similar to mFAS, in contrast to the relatively linear DH domain organization in MAS and modPKSs¹⁵ (Supplementary Fig. 9).

Supplementary Fig. 9 The dimer DH domain organization of LovB compared with its homologs. One domain of the DH dimers from LovB, mFAS, MAS, and CurJ is superimposed. The second domains are shown as ribbons with different colors. The direction of the second domains are indicated with lines.

5. Line 176-178: this sentence is completely unclear and needs rewriting.

A: Basically, we believe that each domain of LovB is relatively stable due to the interaction between KS-DH domain (the “waist”), and not rotated relative with each other, in contrast to the dramatic domain rotation observed in mFAS²⁹.

As suggested, we have now rewrote this sentence in Line 221: “The assembly of two LovB monomers between the KS-DH domains shows the interaction between the upper and lower wings (Supplementary Fig. 7), which could sterically hinder the large-scale domain rotation of LovB, which is in contrast with the observation in mFAS²⁹.”

6. The authors consistently use the term “active sites” where they mean “active site residues”. This should be corrected.

A: We have now corrected the “active sites” to “active site residues” in the text.

7. Line 426: 60.8 e/Å² is the total dose, not the dose rate.

A: This has been corrected to “a total dose of 60.8 e-/Å².” (Line 468)

8. Figure 1e: the lettering of ACP and also LD is too small.

A: The size of ACP and LD labels (Fig. 1e) have been magnified to the same with other domain labels.

9. Figure 3d: The MT domain is labelled “MET” on the figure.

A: We have corrected the “MET” to “CMeT” in Fig. 3d. The “MT” has been revised to “CMeT” throughout the text.

10. Figure 4a: it is unclear where LovB and LovC are in this figure.

A: We have revised Fig.4a to show LovC domain (colored green and transparent).

11. Extended Data Figure 5: this figure is very uninformative. To demonstrate the quality of the 2.9 Å resolution cryo-EM map, please show smaller regions with side chains.

A: We have now shown the representative smaller regions with side chains fitted in the cryo-EM map, next to each domain (Supplementary Fig. 6).

12. Extended Data Table 3: The number of atoms and residues is listed as the same for both structures, but one is labeled LovB and the other LovB with LovC. Either the numbers or the labels must be wrong.

A: We have now corrected the chain number to 4 for EMD-30435 (Supplementary Table 3), with two chains of LovB plus two chains of LovC.

13. Extended Data Table 3: the pixel size is listed as 1 Å, this needs more digits (1.00).

A: The pixel size is revised as 1.00 Å in Supplementary Table 3.

REVIEWERS' COMMENTS

Reviewer #1 (Remarks to the Author, see also attached file):

The authors have generally addressed all comments raised by the reviewers, with only two points remaining:

1. Text on cofactor re-addition, revised line 80: The new statement doesn't include the important statement, that none of the cofactors is observed in the final model. Please add this statement for clarity.

2. Text for structural docking analysis following line 168: The authors only discuss "sample preparation artefacts" as possible causing single-sided binding of LovC, but should also include a moderate overall affinity (association constant) or negative allosteric crosstalk as "physiologically relevant" causes for the observed binding mode of LovC. Please also proof-read new text (LovC felled off).

Additional comment: The combination of close-up maps and clash score provides a better impression of map and model quality and is appreciated. The values for clash score and the observed fit aren't unusual and in line with values obtained in similar studies using standard real space refinement procedures. In the experience of this reviewer, sometimes the automated protocols do fail to provide correct solutions, and require manual fine tuning or correction, often resulting in substantial improvement of clash scores and map fit. It is highly difficult to judge based on the limited data available for review, whether further model optimization is warranted in this case, and this decision is certainly up to the authors. The attached image points to some locations, which might be worth re-inspecting based on the 2D images.

Point-by-point response to the reviewer (second round)

*Reviewer Comments: Black, Helvetica, 9

*Author Responses: Blue, Times New Roman, 10.5

REVIEWER COMMENTS

Reviewer #1

The authors have generally addressed all comments raised by the reviewers, with only two points remaining:

1. Text on cofactor re-addition, revised line 80: The new statement doesn't include the important statement, that none of the cofactors is observed in the final model. Please add this statement for clarity.

A: When we finetuned the model using the ISOLDE, we identified a density that match well with NADP⁺. As suggested, we have now added this statement in Line 89: "None of the cofactors except NADP⁺ was observed in the final models."

Line 148-149 revised to "The NADPH binding pocket with bound NADP⁺ nicotinamide ring can be clearly identified."

2. Text for structural docking analysis following line 168: The authors only discuss "sample preparation artefacts" as possible causing single-sided binding of LovC, but should also include a moderate overall affinity (association constant) or negative allosteric crosstalk as "physiologically relevant" causes for the observed binding mode of LovC. Please also proof-read new text (LovC felled off).

A: Thanks. As suggested, Line 183 and Line 186 have been revised as: ".....some LovC fell off Another possible physiologically relevant cause is that the binding of LovB with LovC has a moderate overall affinity, or there is a negative allosteric crosstalk between them to regulate the synthesis of the polyketide."

Additional comment: The combination of close-up maps and clash score provides a better impression of map and model quality and is appreciated. The values for clash score and the observed fit aren't unusual and in line with values obtained in similar studies using standard real space refinement procedures. In the experience of this reviewer, sometimes the automated protocols do fail to provide correct solutions, and require manual fine tuning or correction, often resulting in substantial improvement of clash scores and map fit. It is highly difficult to judge based on the limited data available for review, whether further model optimization is warranted in this case, and this decision is certainly up to the authors. The attached image points to some locations, which might be worth re-inspecting based on the 2D images.

A: Exactly. We have now manually finetuned the LovB models using the ISOLDE³⁸ software, which is better for EM map and model examination. The clash scores and map fit are improved. The Supplementary Table 3 is updated. The previously unnoticed cofactor NADP⁺ is rebuilt to the KR domain of the models. Figure 3 a and b have been updated.

38. Croll, T. I. ISOLDE: a physically realistic environment for model building into low-resolution electron-density maps. *Acta Crystallogr. D Struct. Biol.* **74**, 519-530 (2018).